# Diagnostic accuracy of direct drug susceptibility testing of second-line antitubercular drugs

Kartiki Srivastava,[1] Varsha Singh,[1] Vinod Kumar Raikwar,[1] Shampa Anupurba[1]

**ABSTRACT** It is well-established that direct drug susceptibility testing (DST) of *Mycobacterium tuberculosis* using a liquid medium for first-line drugs provides accurate and time-saving results. The purpose of this study was to determine whether DST for second-line drugs could be successfully performed using processed smear-positive specimens (direct DST) and whether this method is accurate and may result in a significant reduction in time. The accuracy and shorter turnaround time of this approach were established by comparing the results acquired through direct DST with those obtained through indirect DST. Of the 150 acid-fast bacteria smear-positive sputum specimens that were set up for direct DST, 130 (86.67%) produced results that could be reported. Direct DST reporting took an average of 10 days (range: 9–11 days). The time savings from direct DST to indirect DST, which took into account the time needed to isolate a culture and conduct DST, was 7 days on average (range: 6–9 days). When the direct and indirect DST results were compared, the concordance with levofloxacin (LFX), moxifloxacin (MOX), linezolid (LNZ), and clofazimine (CFZ) were 96.33%, 96.16%, 100%, and 99.24%, respectively. The sensitivity and specificity of the test result were 93.75%, 83.33%, 100%, and 100%, and 98.0, 99.10, 100, and 99.19% with an accuracy of 98%, 98%, 100%, and 99% for LFX, MOX, LNZ, and CFZ, respectively. Direct DST is a fast and accurate diagnostic technique for detecting second-line drug resistance in tuberculosis.

**IMPORTANCE** The significance of this work is that it assesses whether direct drug susceptibility could be used in routine testing to save significant time, which is critical for early diagnosis of resistance and successful treatment.

**KEYWORDS** *Mycobacterium tuberculosis*, drug susceptibility testing, extensively drug-resistance tuberculosis, levofloxacin, moxifloxacin, linezolid, clofazimine

Tuberculosis (TB) is a preventable and usually curable disease. Yet in 2022, TB was the world's second leading cause of death from a single infectious agent, after coronavirus disease (COVID-19), and caused almost twice as many deaths as HIV/AIDS. More than 10 million people continue to fall ill with TB (1). Drug resistance in TB is the biggest global health challenge as it hinders TB control programs and makes the disease worse (2).

For better management of drug-resistant cases, early detection of resistance is extremely important so that effective treatment can be prescribed. Rapid drug susceptibility testing (DST) plays an important role in the detection and control of multidrug-resistant (MDR) TB, which is resistant to both isoniazid (INH) and rifampicin (RIF) drugs, and extensively drug-resistant (XDR) TB, which is MDR TB and is also resistant to any fluoroquinolone and at least one additional Group A drug (3). The drug susceptibility report of antitubercular drugs plays a crucial role in the treatment of disease (4). Recent studies have shown that the mycobacterium growth indicator tube (MGIT960), an automated liquid medium testing method, has become the international gold standard

**Peer Reviewer** Salman H. Siddiqi,, BD Fellow, Sparks, Maryland, USA

Address correspondence to Shampa Anupurba, shampa@bhu.ac.in.

The authors declare no conflict of interest.

for second-line DST of MDR and XDR TB isolates (5–9). Although conventional indirect DST is well established and offers reliable results, the results are typically unavailable within two months after sampling. Long turnaround times (TATs) are not acceptable for patient care, especially in places with high rates of MDR TB (10). Even though liquid-based indirect susceptibility tests have improved the TAT, they are still not rapid enough to allow timely decisions on patient management in the case of MDR and XDR TB. More rapid TB susceptibility tests are needed, particularly in high-TB-burden countries. Recently, the focus has shifted to rapid direct tests in which decontaminated respiratory samples are directly inoculated in drug-free and drug-containing medium or amplified for the detection of MDR and XDR TB (10).

In several studies, direct DST using the BACTEC MGIT 960 system (BD, Sparks, MD, USA) to test RIF and INH susceptibility was found to be highly sensitive and specific and allowed prompt detection of MDR TB (3, 9, 11).

There are no studies on rapid direct drug susceptibility tests for second-line drugs in our knowledge. This study aims to provide a direct drug susceptibility test for second-line drugs and check the accuracy by comparing it with the conventional indirect DST.

## MATERIALS AND METHODS

### Study site and setting

This study was carried out during a period of seven months from November 2023 till May 2024 at the Intermediate Reference Laboratory, Department of Microbiology, Institute of Medical Sciences, Banaras Hindu University, Varanasi, Uttar Pradesh. It was a part of the routine diagnostic workflow under the National TB Elimination Program (NTEP).

### Specimens

Sputum samples from the patients who were positive for MTB by GeneXpert and resistant to either INH or RIF by first-line Line Probe Assay (Genotype MTBDR, Hain LifeScience Germany) were tested for direct DST and were included in this study. Specimens that were found to be smear-positive for acid-fast bacteria (AFB) irrespective of the degree of smear positivity were included in the study. Sputum specimens were transported to the laboratory with minimum delay and were refrigerated if the processing was not done immediately.

### Specimen processing

Specimen processing was done by the standard NALC-NaOH method for digestion, decontamination, and concentration (12). The concentrated sediments were resuspended in about 2–3 mL phosphate buffer (pH 6.8). The decontaminated and concentrated sediments were inoculated into the BACTEC MGIT 960 (BD, USA) automated liquid culture system, used for early detection of mycobacterial growth and drug sensitivity testing (13). A smear was prepared for the acid-fast staining. Before setting up direct DST due to high contamination, a mild redecontamination was done in which NALC was not added and 4% NaOH was added for 8–10 minutes ensuring the bacteria wouldn't get killed.

### Microscopic examination

All the smears were stained with the Ziehl–Neelsen method. Smears were graded based on the number of AFB found during the examination. The smears were graded as scanty (1–9 AFB/100 fields), 1+ (10–99/100 fields), 2+ (1–10 AFB/field), or 3+ (more than 10 AFB/field) (14).

### DNA extraction

DNA extraction was performed by using the GenoLyse kit (Hain Life Science, Germany) according to the manufacturer's instructions. In brief, 1 mL of the decontaminated

sample was centrifuged at 10,000 × $g$ for 15 min. The supernatant was discarded, and the pellet was resuspended into 100 µL lysis buffer (lysA) followed by vortexing for 30 s. This suspension was incubated in the water bath at 95°C for 5 min. Following this, 100 µL neutralization buffer (lysB) was added and vortexed for 30 s, and then it was centrifuged at 15,000 × $g$ for 5 min. The supernatant containing DNA was transferred to a separate tube for further use (15).

## Line probe assay (LPA)

The first-line LPA was carried out with extracted DNA according to the manufacturer's instructions. Results were considered valid only when both amplification control (AC) and conjugate control (CC) probes were present. The probe was only considered present when its intensity of color was similar to or more than that of the AC probe. If all the wild-type probes are present and no mutation probe is present, the sample is considered sensitive (16). As per the NTEP diagnostic algorithm, if samples were found to be resistant to RIF and/or INH, they were further subjected to second-line LPA, and DST was performed for the second-line drugs.

## Direct DST procedure

The direct DST procedure had two major differences from the indirect DST. (i) The control was diluted 1:10 in direct DST while in indirect DST it was diluted 1:100. (ii) In direct DST, an additional antimicrobial mixture of polymyxin B, amphotericin B, nalidixic acid, trimethoprim, and azlocillin (PANTA) (Becton Dickinson Diagnostic Systems, Sparks, MD) was added to the control as well as in the drug-containing MGIT tubes to suppress contamination along with the supplement (3), while in indirect DST, only the OADC supplement was added to the tubes.

All the other reagents and media were the same as those used in the routine indirect DST, i.e., MGIT medium (7 mL bar-encoded MGIT tubes), OADC supplement for DST, and lyophilized drugs moxifloxacin (MOX) (0.25 µg/mL), levofloxacin (LFX) (1 µg/mL), linezolid (LNZ) (1 µg/mL), and clofazimine (CFZ) (1 µg/mL) (Sigma-Aldrich chemical, Ltd, India).

## Preparation of antimicrobial stock solution

MOX, LFX, LNZ, and CFZ drug powder were purchased from Sigma-Aldrich Chemical, Ltd, India. The stock solutions were prepared according to the drug potency using the following formula: (4)

$$\text{Weight (mg)} = \frac{\text{Volume (mL)} \times \text{Concentration } (\mu g/mL) \times (\text{dilution factor})}{\text{Assay potency } (\mu g/mg)}$$

The calculated amount of the drug was dissolved in sterile distilled water (LFX, MOX, LNZ) and dimethyl sulfoxide (CFZ), and an aliquot of the stock solution was used for each test. The stock solution was stored at −20°C.

## Quality control (QC) strain and growth control

*Mycobacterium tuberculosis* H37Rv strain was used for QC testing in LPA and DST. The tubes containing only media, growth supplement, and inoculum were taken as growth control.

## Statistical analysis

Statistical analysis was done by using the online statistical calculator Med calc. Sensitivity, specificity, negative predictive value (NPV), and positive predictive value (PPV) with 95% confidence intervals were calculated. The Kappa statistic (κ) was used to calculate the agreement between the direct and indirect DST. Test reliability measured by the κ

value was interpreted as follows: <0.2 poor; 0.21–0.4 fair; 0.41–0.6 moderate; 0.61–0.8 good; and ≥0.81 excellent (17). Indirect DST was taken as the gold standard.

## RESULTS

A total of 5,075 sputum samples were detected positive for MTB from GeneXpert from the period of Nov 2023 till May 2024.

All the 5,075 positive samples, despite being RIF resistant or sensitive in GeneXpert, were tested for first-line LPA. 576 samples were detected as RIF-resistant, MDR, or INH mono-resistant (Table 1) and were AFB smear positive. Direct DST was set up for 150 samples.

In Table 2, the average time to report direct DST was found to be 10 days. The time was calculated according to smear-positive grading. The majority of the specimens were 2 to 3+. Of 150 specimens, 9 specimens had ×200 errors where the control did not reach the required threshold, which is due to low bacterial load, and 11 specimens had ×400 errors due to contamination of other microorganisms.

### Comparison of direct and indirect DST

Direct and indirect DST was performed on 150 samples. The overall reportable direct DST was 130 (86.67%) out of 150 samples. While in indirect DST, it was 136 (90.67%). 20 DST (13.33%) were not reportable in direct DST, and 14 (6.67%) were not reportable in indirect DST. Discrepant results were analyzed on all direct DST in which confirmed indirect DST was available (Table 3). Of 130 specimens, 10 (7.69%) specimens showed discrepant results between direct and indirect DST.

### Statistics

An excellent agreement was found between the two tests with the κ values of 0.884, 0.943, 1.0, and 0.843 for LFX, MOX, LNZ, and CFZ, respectively. In this report, we got a sensitivity of 93.75%, 83.33%, 100%, and 100% and specificity of 98.0%, 99.10%, 100%, and 99.19% with an accuracy of 98%, 98%, 100%, and 99% for LFX, MOX, LNZ, and CFZ, respectively (Table 4).

## DISCUSSION

Drug resistance in TB has become a significant problem with an urgent need for rapid detection of drug-resistant methods. The direct DST method plays an important role in saving time and early detection of drug resistance in TB. Commercially available molecular assays, such as Genotype MTBDR (Hain Lifescience, Germany), can be applied directly to smear-positive specimens and have less TAT (18). However, for the testing of second-line drugs, it only targets fluoroquinolones and second-line injectable antituberculosis drug groups. None of the established molecular tests target all possible genes involved in resistance, and thus a variable proportion of resistant strains may not be detected (19). There have been many studies on direct DST for first-line drugs (9–11), but to the best of our knowledge, there is no study conducted on direct DST for second-line drugs. Our main objective was to perform direct DST for second-line drugs and make it rapid as well as cost-effective.

In previous studies, the protocol for the direct DST was extended from 4–13 days to 4–21 days because the bacterial count present in the inoculum was low in comparison

**TABLE 1** Distribution of INH mono RIF-resistant and MDR samples through LPA

| Description | No. of samples |
| --- | --- |
| INH mono-resistant | 281 |
| RIF resistant | 20 |
| RIF + INH resistant | 275 |
| Total | 576 |

**TABLE 2** Overall summary of testing

| Smear grading | DST setup time (in Days) | | | Reportable DST | ×200 error | ×400 error |
|---|---|---|---|---|---|---|
| | 4–7 Days | 8–13 Days | >13 Days | | | |
| Scanty (26) | 4 | 7 | 15 | 19 | 5 | 2 |
| 1+ (34) | 10 | 20 | 4 | 28 | 2 | 4 |
| 2+ (39) | 16 | 22 | 1 | 35 | 1 | 3 |
| 3+ (51) | 21 | 30 | 0 | 48 | 1 | 2 |
| Total (150) | 51 | 79 | 20 | 130 | 09 | 11 |

to the culture isolates (3). Since the protocol of 21 days is for pyrazinamide, the setup was of two tubes, one for GC and one for the drug, resulting in high cost because a growth control was needed for each drug. In this study, we did not extend the protocol to 21 days and set up with a regular 4–13-day protocol. As a result, some tests showed ×200 error due to less bacterial count, but when observed visually, the granular and flaky appearance of MTB was seen. In those cases, the tubes were incubated at 37°C for one week, and then the interpretation was done manually. For the confirmation of MTB, the capilia and smear microscopy were done. If the proper growth was not visible even after one week of manual incubation, then the test was considered a ×200 error. Nineteen out of 24 scanty-positive samples could be reported. The ×200 error was observed in five scanty-positive specimens, 2 with 1+, 1 with 2+, and 1 with 3+ grading.

The smear-positive specimen had an acceptable incidence of contamination (4 to 8%) and a very high culture positivity rate (above 95%). DST from smear-positive specimens had an overall success rate of 86.67%. In other words, only roughly <14% of all DST setups were unreportable for a variety of reasons, including contamination (×400 errors) or insufficient control growth (×200 errors). There was no nontuberculous mycobacteria species identified in our study.

Saving time was the primary objective of this study. Reporting Bactec MGIT indirect DST from cultures that tested positive took anywhere from six to thirteen days (20). It is common knowledge that resistant isolates require more time to process than susceptible ones. Since the positive culture needed 10–12 days to grow, it took 18–20 days to report the indirect DST. However, for reporting direct DST, we only needed 10–12 days. Direct DST reporting began as soon as the processed specimen was inoculated and the desired DST result was obtained. It required two to three days to screen for first-line drug resistance because this study focused on second-line drugs. GeneXpert and Genotype MTBDRplus (Hain Lifescience, Germany) were used to screen for first-line drug resistance, which took 1–2 days, and the resistant strains were selected for the second-line DST. In total, it took an average of 22 days to report Indirect DST, while to report Direct DST, it only took an average of 15 days (Fig. 1). Due to this time lag between sample processing and selection of first-line resistant strains, the chances of contamination were high, and to prevent that a mild redecontamination procedure was done in which decontamination was done with 4% NaOH without the addition of NALC and was left for 8–10 minutes. The rest of the procedure was the same as the standard decontamination process. These changes were made because many studies have shown that twice decontamination lowers the mycobacterial load (21).

**TABLE 3** Comparison of discrepant results of direct and indirect DST (levofloxacin, moxifloxacin, linezolid, and clofazimine)

| Drugs | No. (%) of specimens | | |
|---|---|---|---|
| | False S | False R | Total |
| Levofloxacin | 2 (1.53) | 2 (1.53) | 4 (3.07) |
| Moxifloxacin | 4 (3.07) | 1 (0.76) | 5 (3.84) |
| Linezolid | 0 | 0 | 0 |
| Clofazimine | 0 | 1 (0.76) | 1 (0.76) |
| Total | 6 (4.6) | 4 (3.07) | 10 (7.69) |

**TABLE 4** Sensitivity, specificity, PPV, NPV, and accuracy of LFX, MOX, LNZ, and CFZ were calculated with 95% CI

| Statistics | LFX | MOX | LNZ | CFZ |
|---|---|---|---|---|
| Sensitivity | 93.75 | 83.33 | 100 | 100 |
| Specificity | 98.00 | 99.10 | 100 | 99.19 |
| PPV | 93.57 | 95.24 | 100 | 85.71 |
| NPV | 98.00 | 96.49 | 100 | 100 |
| Accuracy | 98.00 | 98.00 | 100 | 99.00 |

Another important aspect of the current findings is the accuracy of the direct DST method. In this study, we got a high concordance rate with the indirect DST, which is considered the gold standard. According to Siddiqi et al., the results of direct DST were compared with those of indirect DST; there was 95.1% concordance with INH and 96.1% with RIF (3). Likewise, another study has reported that direct DST is highly sensitive and reliable when compared with indirect DST (11). In this study, the concordance of direct DST compared to indirect DST was 96.93% with LFX, 96.16% with MOX, 100% with LNZ, and 99.24% with CFZ. Other than the proportional method, there were studies on direct DST with NRA methods for which the sensitivity and specificity reported by Gupta et

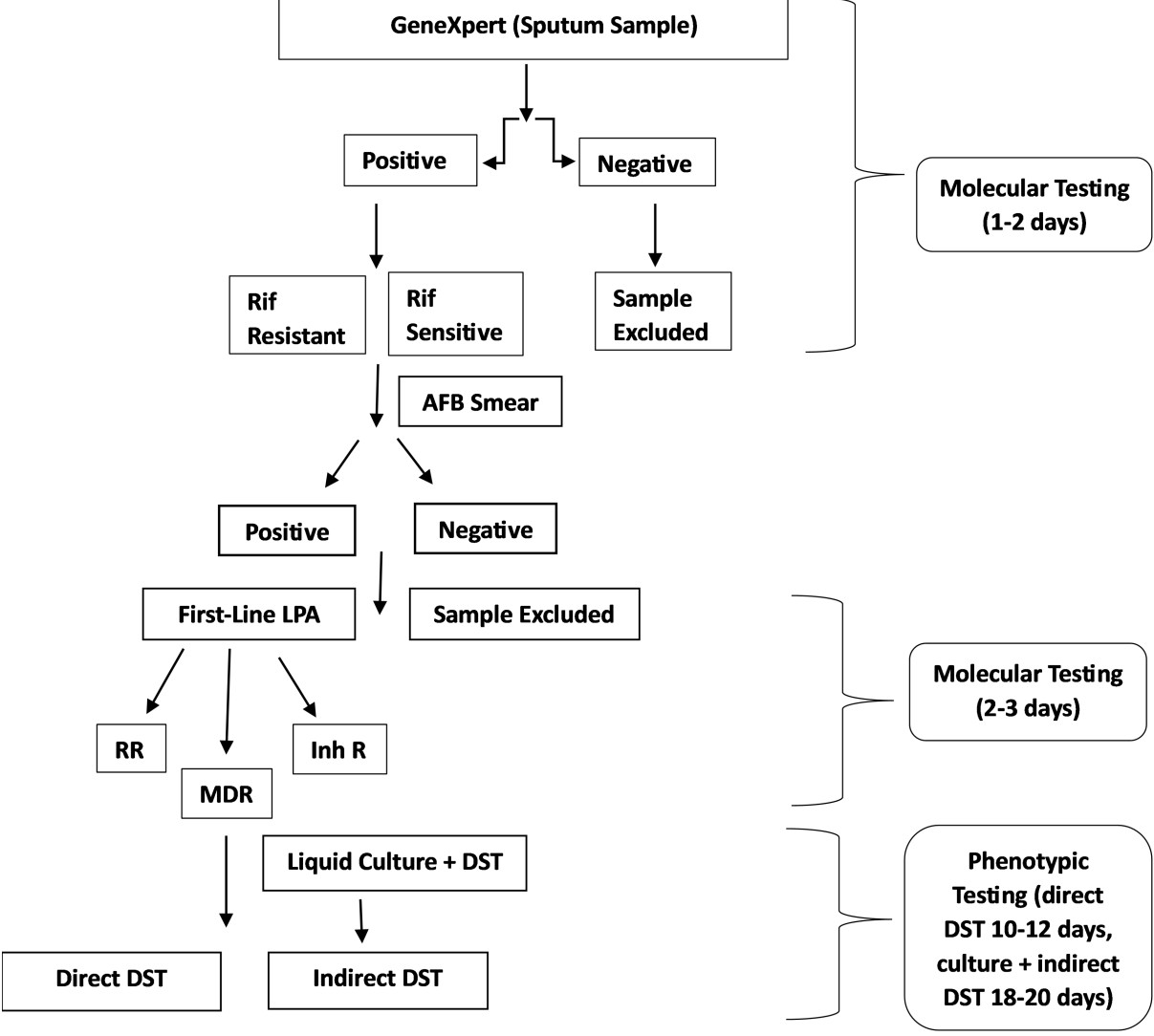

**FIG 1** Schematic representation of the Workflow of Direct and Indirect DST

al. (17) were 98.4%, 97%, 88.5%, and 94.2% and 100%, 100%, 94%, and 99% for RIF, INH, streptomycin (STR), and ethambutol (EMB), respectively (17). In another study, the sensitivity and specificity reported of direct DST NRA methods were 90% and 97.3%, 92.6% and 98.2%, 52.9% and 100%, and 28.6% and 100% for RIF, INH, STR, and EMB, respectively (22). In this study, we got sensitivity of 93.75%, 83.33%, 100%, and 100% and specificity of 98.0%, 99.10%, 100%, and 99.19% with an accuracy of 98%, 98%, 100%, and 99% for LFX, MFX, LNZ, and CFZ, respectively.

The primary drawback of this study is that the sample was kept for 3–4 days, increasing the likelihood of contamination. As a result, a light redecontamination was performed before the DST was set up, requiring more personnel. In addition, the bacterial count was reduced as a result of the double decontamination due to this DST for specimens that have scanty smear grading should be performed using an indirect DST method.

Our study concludes that direct DST is a reliable and time-saving diagnostic method for the detection of second-line drug resistance in TB. Despite direct DST having a requirement of more manpower than indirect DST, the findings of the direct DST test showed excellent agreement ranging between 84.3% and 100% with the results of the indirect method, proving that direct DST is an accurate and reliable test.

## ACKNOWLEDGMENTS

We thank the NTEP Ministry of Health and Family Welfare Government of India for their valuable support to this study. The current study was not supported by any funding agency.

K.S. and S.A. designed the study. K.S., V.S., and V.K.R. executed the work and collected the data. K.S., V.S., V.K.R., and S.A. interpreted the results. K.S. primarily wrote the manuscript. S.A. provided valuable insight for revising the manuscript. All authors approved the final manuscript.

## AUTHOR AFFILIATION

[1]Department of Microbiology, Institute of Medical Sciences, Banaras Hindu University, Varanasi, Uttar Pradesh, India

## AUTHOR ORCIDs

Shampa Anupurba http://orcid.org/0000-0002-7644-1086

## AUTHOR CONTRIBUTIONS

Kartiki Srivastava, Formal analysis, Methodology, Visualization, Writing – original draft | Varsha Singh, Formal analysis, Methodology, Visualization | Vinod Kumar Raikwar, Methodology, Visualization | Shampa Anupurba, Conceptualization, Supervision, Writing – review and editing

## ETHICS APPROVAL

This study has been ethically approved by the Institute ethical committee of the Institute of Medical Sciences (ECR/526/Inst/UP/2014), Banaras Hindu University, Varanasi.

## ADDITIONAL FILES

The following material is available online.

Open Peer Review

**PEER REVIEW HISTORY (review-history.pdf).** An accounting of the reviewer comments and feedback.

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
