## [Reviewer comments · Microbiology Spectrum]

Microbiology Spectrum

Diagnostic Accuracy of Direct Drug Susceptibility Testing of Second-line Anti-tubercular drugs

Kartiki Srivastava, Varsha Singh, Vinod Raikwar, and Shampa Anupurba

Corresponding Author(s): Shampa Anupurba, Banaras Hindu University Institute of Medical Sciences

Review Timeline:

Submission Date:	October 3, 2024
Editorial Decision:	January 27, 2025
Revision Received:	February 6, 2025
Accepted:	February 24, 2025

Editor: Michael Whitfield

Reviewer(s): Disclosure of reviewer identity is with reference to reviewer comments included in decision letter(s). The following individuals involved in review of your submission have agreed to reveal their identity: Salman H. Siddiqi (Reviewer #2)

Transaction Report:

DOI: <https://doi.org/10.1128/spectrum.02506-24>

Re: Spectrum02506-24 (Diagnostic accuracy of direct drug susceptibility testing of second-line anti-tubercular drugs)

Dear Prof. Shampa Anupurba:

Thank you for the privilege of reviewing your work. Below you will find my comments, instructions from the Spectrum editorial office, and the reviewer comments.

Please carefully revise the manuscript with the reviewer comments in mind. In addition, please edit for consistency throughout the manuscript and where possible condense to make it clear to the readers.

Revision Guidelines

Sincerely,
Michael Whitfield
Editor
Microbiology Spectrum

Reviewer #1 (Comments for the Author):

- Please be consistent with capitalization throughout (for example Acid-fast, drug names such as linezolid)
- Please be consistent with abbreviations throughout (for example TB, tuberculosis, SLID, spelling out numbers versus using numerals)

- In abstract, rephrase "The statistical study revealed" to simply state the sensitivity and specificity of the test results
- In importance, suggest removing " The significance of this work is that" and replacing with "This study"
- Introduction, paragraph 2- MDR and XDR need to be spelled out in first use. Furthermore, suggest defining these terms.
- Introduction, paragraph 2- "This is such a long time that it can hardly be seen as ..." revise sentence to state that "Long turnaround times are not acceptable for patient care, especially in places with high rates of MDR-TB."
- Study site and setting- spell out "November"
- Specimens- please provide details of which line probe assay was utilized
- Line probe assay- use acronym after spelling out in paragraph
- Line probe assay- spell out "mutation" for "mut"
- State antibiotic abbreviation in "Direct DST" section and then use throughout
- Results, paragraph 3- were all 9 X200 errors due to low abundance (ie scant)? What contaminants are observed for the X400 errors- other mycobacteria or other microorganisms?
- Discussion, paragraph 3- Instead of estimating 10-15% unreportable, state <14% were unreportable since this can be calculated.

-

Reviewer #2 (Comments for the Author):

The time frame and workflow are not clear regarding molecular testing and then performing direct DST along with culture isolation and the indirect DST. For statiistical analysis did the indirect DST results were taken as the standard? What is the recommendation for a more simple and more economical approach for high-burden countries where it is needed the most? In Siddiqi et al study they suggested to incubate the tubes which were negative with DST protocol further with culture protocol in MGIT so that the direct DST could be manually interpreted with a longer time. Is it possible to use manual reader if it is still available.

Response to Reviewers

We sincerely thank the esteemed reviewers for their valuable and insightful comments, which have significantly contributed to enhancing the quality of our manuscript. We have carefully considered each suggestion and made the necessary revisions to address the identified concerns. Our team has made every effort to rectify any flaws and ensure clarity, coherence, and accuracy throughout the manuscript. We greatly appreciate the reviewers' dedication and expertise in improving our work.

Reviewer's Comment/Suggestion	Action Taken/Response
Reviewer #1	
Please be consistent with capitalization throughout (for example Acid-fast, drug names such as linezolid)	Thank you for pointing this out. We have made the necessary changes.
Please be consistent with abbreviations throughout (for example TB, tuberculosis, SLID, spelling out numbers versus using numerals)	We agree with this and have incorporated your suggestion throughout the manuscript.
In abstract, rephrase 'The statistical study revealed' to simply state the sensitivity and specificity of the test results.	We appreciate your suggestion and have rephrased the sentence accordingly.
In importance, suggest removing 'The significance of this work is that' and replacing it with 'This study'	Revision done.
Introduction, paragraph 2 - MDR and XDR need to be spelled out in first use. Furthermore, suggest defining these terms.	Thank you for pointing this out. We have made the appropriate changes.
Introduction, paragraph 2 - 'This is such a long time that it can hardly be seen as ...' revise sentence to state that 'Long turnaround times are not acceptable for patient care, especially in places with high rates of MDR-TB.'	Revision done.
Study site and setting - spell out 'November'	Incorporated.
Specimens - please provide details of which line probe assay was utilized	We have provided the details.
Line probe assay - use acronym after spelling out in paragraph	Correction made throughout the manuscript.
Line probe assay - spell out 'mutation' for 'mut'	Revision made.
State antibiotic abbreviation in 'Direct DST' section and then use throughout	Thank you for the suggestion we have made the changes throughout the

	manuscript.
Results, paragraph 3 - were all 9 X200 errors due to low abundance (i.e., scant)? What contaminants are observed for the X400 errors - other mycobacteria or other microorganisms?	(i) Yes, all 9 X200 were due to low abundance of bacteria. (ii) Other microorganisms were observed (not NTM) for the X400 error. We did not specify it further.
Discussion, paragraph 3 - Instead of estimating 10-15% unreportable, state <14% were unreportable since this can be calculated.	Suggestion is appreciated and correction has been made accordingly.
Reviewer #2	
The time frame and workflow are not clear regarding molecular testing and then performing direct DST along with culture isolation and the indirect DST. For statistical analysis, did the indirect DST results were taken as the standard?	(i) Thank you for pointing this out. We have incorporated a picture of the workflow and time frame of the work and also discussed it. (ii) Yes, For the statistical analysis indirect DST was taken as the gold standard.
What is the recommendation for a more simple and more economical approach for high-burden countries where it is needed the most? In Siddiqi et al.'s study, they suggested incubating the tubes which were negative with the DST protocol further with culture protocol in MGIT so that the direct DST could be manually interpreted with a longer time. Is it possible to use a manual reader if it is still available?	(i) You have raised an important point here. According to us for testing susceptibility to second-line drugs in HBC, direct DST is best for a simple and economical approach. (ii) The manual reading according to Siddiqi et al. is a good economical approach but since the direct DST for SL drug is already a laborious process, we think this approach can be used for first-line direct DST only.

Re: Spectrum02506-24R1 (Diagnostic Accuracy of Direct Drug Susceptibility Testing of Second-line Anti-tubercular drugs)

Dear Prof. Shampa Anupurba:

Your manuscript has been accepted, and I am forwarding it to the ASM production staff for publication. Your paper will first be checked to make sure all elements meet the technical requirements. ASM staff will contact you if anything needs to be revised before copyediting and production can begin. Otherwise, you will be notified when your proofs are ready to be viewed.

Sincerely,
Michael Whitfield
Editor
Microbiology Spectrum